# How Distance Transform Maps Boost Segmentation CNNs: An Empirical Study

**Jun Ma**[1]                                                                    JUNMA@NJUST.EDU.CN
[1] *Department of Mathematics, Nanjing University of Science and Technology*
**Zhan Wei**[2]                                                                  WEIZHAN@HDU.EDU.CN
[2] *HangZhou DianZi University*
**Yiwen Zhang**[3]                                                              WHISNEY@I.SMU.EDU.CN
[3] *School of Biomedical Engineering, Southern Medical University*
**Yixin Wang**[4]                                                         WANGYIXIN19@MAILS.UCAS.AC.CN
[4] *Institute of Computing Technology, Chinese Academy of Sciences; University of Chinese Academy of Sciences*
**Rongfei Lv**[5]                                                               LRF@CQU.EDU.CN
[5] *College of Optoelectronic Engineering, Chongqing University*
**Cheng Zhu**[6]                                                               ZHUC@HCSD-MED.COM
[6] *Shenzhen Haichuang Medical Co. Ltd*
**Gaoxiang Chen**[*7]                                                        GAOXIANGCHEN@WMU.EDU.CN
[7] *The First Affiliated Hospital of Wenzhou Medical University*

**Jianan Liu**[*8]                                                          LJNNZB@MAIL.NANKAI.EDU.CN
[8] *College of Artificial Intelligence, Nankai University*

**Chao Peng**[*9]                                                           CQUPENGCHAO@CQU.EDU.CN
[9] *College of Optoelectronic Engineering, Chongqing University*

**Lei Wang**[*10]                                                           WANGLEI_NUIST@126.COM
[10] *School of Automation, Nanjing University of Information Science and Technology*
**Yunpeng Wang**[11]                                                       18111510027@FUDAN.EDU.CN
[11] *Institutes of Biomedical Sciences, Fudan University*
**Jianan Chen**[12]                                                         CHENJN2010@GMAIL.COM
[12] *Department of Medical Biophysics, University of Toronto*

## Abstract

Incorporating distance transform maps of ground truth into segmentation CNNs has been an interesting new trend in the last year. Despite many great works leading to improvements on a variety of segmentation tasks, the comparison among these methods has not been well studied. In this paper, our *first contribution* is to summarize the latest developments of these methods in the 3D medical segmentation field. The *second contribution* is that we systematically evaluated five benchmark methods on two representative public datasets. These experiments highlight that all the five benchmark methods can bring performance gains to baseline V-Net. However, the implementation details have a noticeable impact on the performance, and not all the methods hold the benefits on different datasets. Finally, we suggest the best practices and indicate unsolved problems for incorporating distance transform maps into CNNs, which we hope would be useful for the community. The codes and trained models are publicly available at https://github.com/JunMa11/SegWithDistMap.

---

* Contributed equally

**Keywords:** Distance transform maps, medical image segmentation, convolutional neural networks, signed distance function

## 1. Introduction

Convolutional neural networks (CNNs)[1] have been widely used on a variety of medical image segmentation tasks, and achieved great success, such as liver segmentation (Bilic et al., 2019), heart segmentation (Bernard et al., 2018), brain segmentation (Wang et al., 2019b) and so on. Recently, a new segmentation methodology is emerging where the distance transform maps are incorporated into existing CNNs (Kervadec et al., 2019; Karimi and Salcudean, 2019; Xue et al., 2020; Navarro et al., 2019; Dangi et al., 2019) to obtain further improvements.

Most existing CNNs use binary or multi-label mask as ground truth. Distance transform maps (DTM) offer an alternative to classical ground truth. For example, a binary mask can be transformed to a graylevel image, termed as distance transform map, where the intensities of pixels in the foreground are changed according to the distance to the closest boundary. One can also compute the signed distance function (SDF) of the ground truth, which embeds object contours in a higher dimensional space. In general, the signed distance function takes negative values inside the object and positive values outside the object. The absolute value is defined by the distance between the point of interest and the closest boundary point. In a word, distance transform map or signed distance function is an implicit representation of ground truth, and there exists a rigorous mapping between them.

In the past year, incorporating the distance transform maps of image segmentation labels into CNNs pipelines has received significant attention. These methods can be classified into two classes (Figure 1) in terms of the usage of distance transform maps: (1) new loss functions (Kervadec et al., 2019; Karimi and Salcudean, 2019; Xue et al., 2020): use distance transform maps to design new loss functions; and (2) adding auxiliary tasks (Navarro et al., 2019; Dangi et al., 2019): generating the segmentation probabilistic map and regressing the distance transform maps at the same time.

All these methods argue that using distance transform maps can boost existing baseline CNNs, such as U-Net and V-Net. However, these methods are tested on different datasets, and there is no shared experimental protocol followed by all. Thus, we do not know which method should be chosen to improve performance in practice.

This paper aims to experimentally answer the question:

*How can distance transform maps boost segmentation CNNs?*

Our contributions are summarized as follows:

- summarizing the latest developments about incorporating distance transform maps into CNN-based 3D medical image segmentation.

- benchmarking five methods on two representative datasets by extensive experiments.

---

1. In this paper, CNNs refers specifically to the networks for medical image segmentation.

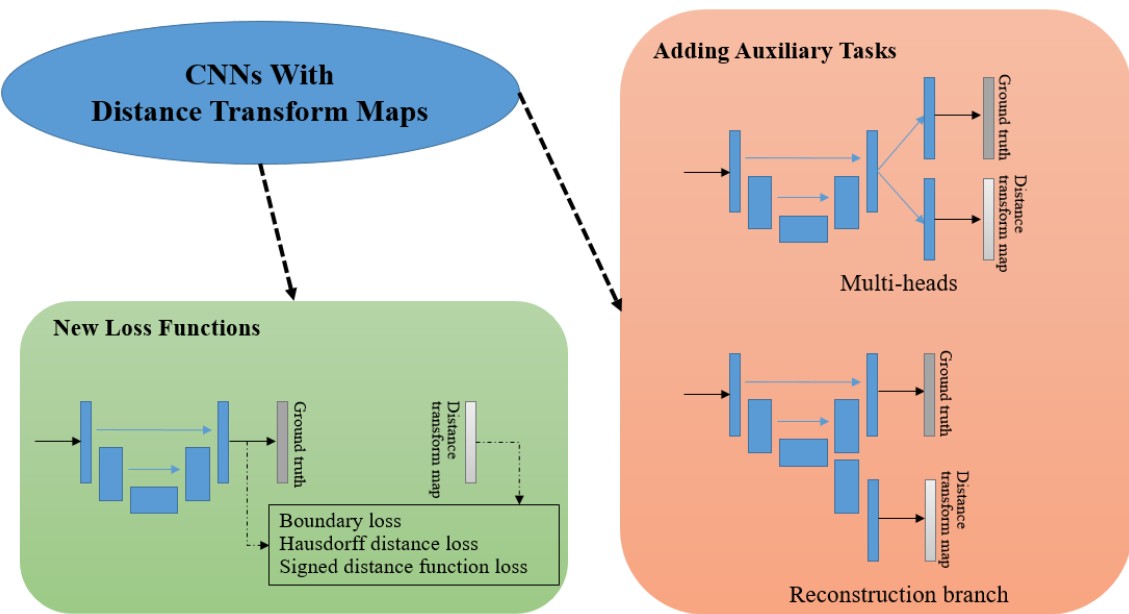

Figure 1: Overview of the two categories of recent distance transform maps-related CNNs in medical image segmentation.

The rest of the paper is organized as follows. A brief review of the recent[2] DTM-related CNNs in 3D medical image segmentation is given in Section 2. We present the experimental settings in Section 3 and the corresponding results in Section 4. Finally, we conclude this paper in Section 5.

## 2. CNNs with Distance Transform Maps

In this section, we present an overview of five benchmark methods that are selected based on two criteria: (1) the method is general and can be applied to many 3D segmentation tasks; (2) The method is published in 2019-2020. Several related methods also use distance transform maps, but they are designed for specific tasks such as tubular segmentation (Wang et al., 2019a) and lesion detection (van Wijnen et al., 2019). Evaluating these tailored methods is beyond the scope of this paper.

### 2.1. Basic Notation

Let $\Omega$ denote the grid on which the image $I$ is defined, and $G, S$ denote the corresponding ground truth and segmentation, respectively. $S_\theta$ denotes the softmax outputs of CNNs where $\theta$ is the parameters. Formally, we define the distance transform map (DTM) of

---

2. Here, "recent" means after 2019 in this paper.

ground truth $G$ by

$$G_{DTM} = \begin{cases} \inf_{y \in \partial G} ||x - y||_2, & x \in G_{in} \\ 0, & others \end{cases} \tag{1}$$

where $||x - y||_2$ is the Euclidian distance between voxels $x$ and $y$, and $G_{in}$ denotes the inside of the object. The signed distance function (SDF) of ground truth $G$ is defined by

$$G_{SDF} = \begin{cases} -\inf_{y \in \partial G} ||x - y||_2, & x \in G_{in} \\ 0, & x \in \partial G \\ \inf_{y \in \partial G} ||x - y||_2, & x \in G_{out} \end{cases} \tag{2}$$

where $G_{out}$ and $\partial G$ denote the outside and boundary of the object, respectively. The main difference between distance transform map $G_{DTM}$ and signed distance function $G_{SDF}$ is that the $G_{SDF}$ considers the distance transformation information of both foreground and background, while $G_{DTM}$ only computes the distance transformation of the foreground.

In the following two subsections, we give a brief review of five methods that will be evaluated in Section 3. As shown in Figure 1, we divided the five methods into two categories, new loss functions and adding auxiliary tasks, based on their main contributions (the usage of distance transform maps).

## 2.2. New Loss Functions

Kervadec et al. (Kervadec et al., 2019) proposed boundary loss (BD) to mitigate unbalanced segmentation problems. The key idea is to use an integral approach for computing boundary variations between segmentation and ground truth, which avoids complex local differential computations. Specifically, the loss is defined by:

$$L_{BD} = \frac{1}{|\Omega|} \sum_{\Omega} G_{SDF} \circ S_\theta \tag{3}$$

where $G_{SDF}$ denotes the signed distance function of ground truth $G$, and $\circ$ is the Hadamard (i.e. voxel-wise) product.

To reduce the Hausdorff distance (HD) during training CNNs, Karimi et al. (Karimi and Salcudean, 2019) proposed Hausdorff distance loss for direct minimization of HD. The loss function is defined by

$$L_{HD} = \frac{1}{|\Omega|} \sum_{\Omega} [(S_\theta - G)^2 \circ (G_{DTM}^2 + S_{DTM}^2)] \tag{4}$$

where $G_{DTM}$ and $S_{DTM}$ denote the distance transform maps of ground truth $G$ and predicted segmentation $S$, respectively.

Recently, Yuan et al. (Xue et al., 2020) proposed using CNNs to directly regress the signed distance function (SDF) of ground truth rather than to generate softmax outputs, because there is rigorous mapping between the ground truth and the SDF. The signed distance function regression loss is defined by

$$L_{SDF} = -\sum_{\Omega} \frac{G_{SDF} \circ S_{SDF}}{G_{SDF}^2 + S_{SDF}^2 + G_{SDF} \circ S_{SDF}} \tag{5}$$

where $G_{SDF}$ and $S_{SDF}$ denote the ground truth and the predicted signed distance functions, respectively. The SDF loss aims to penalize the output SDF with wrong sign.

In summary, the distance transform map (DTM) of ground truth was incorporated in all the three loss functions. Boundary loss (Kervadec et al., 2019) assigned weights to the softmax probability outputs based on the ground truth SDF, while Hausdorff distance loss (Karimi and Salcudean, 2019) introduced not only the ground truth DTM but also the predicted segmentation DTM to weight the softmax probability outputs. SDF loss (Xue et al., 2020) employed the product of predicted SDF and ground truth SDF to guide the SDF regression network during training.

In practice, it should be noted that the three loss functions should be coupled with Dice loss so as to stabilize training process, especially at the beginning of training, otherwise training may not converge. More details about the usage of the loss functions are presented in Section 3.2.

### 2.3. Auxiliary Tasks

Distance transform maps can also be used to augment CNNs by adding auxiliary tasks. Usually, the auxiliary task is a regression task, and we found two different ways to regress the DTM from recent publications. First, a new head sharing the same backbone network can be added to the end of the CNNs (Figure 1, top right), for the purpose of learning shape information of chest organs (Navarro et al., 2019) or tubular structure reconstruction (Wang et al., 2019c). The other way is to add a reconstruction branch for learning robust global features by regressing pixel-wise distance map (Figure 1, bottom left) (Dangi et al., 2019).

In summary, both multi-heads and reconstruction-branch CNNs aim at regressing the DTM of ground truth. The main difference is that the multi-heads CNN shares the backbone network while the reconstruction-branch CNN only shares the encoder network. In addition, we observed that these methods only consider the DTM of foreground, but not the SDF of ground truth which consists of the DTMs of both foreground and background. To the best of our knowledge, regressing the SDF of ground truth has not been explored in existing studies.

## 3. Experiments

In this section we describe the datasets, the backbone CNN, quantitative segmentation metrics and experimental design.

### 3.1. Dataset, network backbone, and metrics

We use two representative datasets to evaluate the above five benchmark methods. One dataset is the left atrial (LA) MRI, which is an organ segmentation task[3]. The other dataset is the liver tumor CT (LiTS) ,which is a popular tumor segmentation task[4]. LA includes 100 3D gadolinium-enhanced MR training cases. We randomly selected 16 cases for training and 20 cases for testing to create a typical small sample learning setting. LiTS includes 118

---

3. MICCAI 2018 left atrial segmentation: http://atriaseg2018.cardiacatlas.org/.

4. MICCAI 2017 liver tumor segmentation: https://competitions.codalab.org/competitions/17094

CT training cases. We split them into 90 for training and 28 for testing. All the cases were cropped centering at the heart or liver region for better comparison of the segmentation performance of different methods, and normalized by subtracting the mean and divided by standard deviation. We chose these two datasets because we want to involve typical modalities (CT and MR), tasks (organ and tumor) and challenges (small sample learning and small objects segmentation) in 3D medical image segmentation tasks.

We employ V-Net (Milletari et al., 2016) as the network backbone. It has five stage convolutional blocks in different resolutions. The base convolution block ($1_{st}$ stage) has 16 feature maps, and the number of feature maps is doubled every next stage. During training, we used the Adam optimizer for all experiments and searched the best leaning rate in the set $\{0.01, 0.001, 0.0001\}$. To make the experiments reproducible, we set the random seed as 2019. We also added two dropout layers after the $L - 5_{th}$ and $R - 1_{st}$ stage layers[5] with dropout rate 0.5. For left atrial MRI dataset, dropout was turned on during training, but turned off during inference. Using dropout could bring performance gains on left atrial MRI dataset. However, we fould that using dropout hurts the performance on liver tumor CT dataset based on our experiments. Hence, we turned off dropout in all experiments for liver CT tumor segmentation. All the networks and loss functions are implemented in PyTorch, and run in Linux.

Four complementary segmentation metrics are introduced to quantitatively evaluate the segmentation results. Dice and Jaccard, two region-based metrics, are used to measure the region mismatch. Average surface distance (ASD) and 95% Hausdorff Distance (95HD), two boundary-based metrics, are used to evaluate the boundary errors between the segmentation results and the ground truth.

### 3.2. Experimental design

We evaluated the five benchmark methods on the two representative datasets with the above training protocol. For boundary loss and Hausdorff distance loss, the final loss function is defined by

$$L = \alpha L_{Dice} + (1 - \alpha)L(\cdot) \tag{6}$$

where $\alpha \in [0, 1]$ is the weight parameter, and $L(\cdot)$ denotes boundary loss and Hausdorff distance loss, respectively. In practice, $\alpha$ is set to 1 at the start of the training and decreased by 0.001 after each epoch until it reaches 0.01, which is suggested in (Kervadec et al., 2019; Karimi and Salcudean, 2019). For signed distance function loss, the final loss is defined by

$$L = L_{Dice} + 10(L1 + L_{SDF})$$

as suggested in (Xue et al., 2020). In addition, For multi-heads and reconstruction-branch CNNs, directly regressing the signed distance function is still undeveloped as we mentioned in Section 2.3. Thus, we also evaluated several combinations among different network architectures (multi-heads versus reconstruction-branch CNNs), different regression tasks (DTM versus SDF), and different loss functions ($L1$, $L2$ or $L1 + L2$).

---

5. L and R denote the left encode path and right decode path in V-Net.

## 4. Results and Discussion

In this section, we present the quantitative results of the five benchmark methods on the two datasets.

Table 1: Quantitative results with average (standard deviation) on left atrial MRI segmentation. FG, DTM and SDF denote the foreground distance transform map and the signed distance function, respectively. Rec-Branch denotes the the network with reconstruction branch, and L1/L2 denotes . The arrows indicate which direction is better.

| Methods | Dice (%) ↑ | Jaccard (%) ↑ | 95HD ↓ | ASD ↓ |
|---|---|---|---|---|
| V-Net baseline | 84.4 (5.70) | 73.6 (7.00) | 20.1 (13.8) | 5.29 (3.43) |
| Boundary loss | 85.0 (5.64) | 74.2 (7.87) | 20.8 (15.0) | 5.43 (3.43) |
| Hausdorff distance loss | **85.5 (4.96)** | **75.0 (7.30)** | 15.9 (13.3) | 4.46 (3.68) |
| Signed distance function loss | 84.2 (8.48) | 73.5 (11.0) | **13.5 (11.2)** | **3.24 (3.10)** |
| Multi-heads: FG DTM-L1 | 83.7 (6.33) | 72.5 (8.97) | 24.7 (12.8) | 6.62 (3.32) |
| Multi-heads: FG DTM-L2 | 82.6 (6.87) | 71.0 (9.65) | 15.5 (11.5) | 4.10 (3.12) |
| Multi-heads: FG DTM-L1+L2 | 83.3 (10.7) | 72.6 (12.6) | 17.5 (12.1) | 4.87 (3.12) |
| Multi-heads: SDF-L1 | 85.5 (7.82) | 75.3 (10.2) | **11.8 (8.86)** | **2.65 (2.11)** |
| Multi-heads: SDF-L2 | **87.0 (3.49)** | **77.2 (5.49)** | 16.1 (13.5) | 3.97 (3.14) |
| Multi-heads: SDF-L1+L2 | 84.5 (4.38) | 73.5 (6.49) | 24.7 (15.0) | 6.09 (3.71) |
| Rec-Branch: FG DTM-L1 | 83.5 (5.91) | 72.2 (8.30) | 23.6 (14.8) | 5.45 (3.57) |
| Rec-Branch: FG DTM-L2 | 81.5 (8.40) | 69.5 (10.9) | 19.5 (16.9) | 4.49 (4.76) |
| Rec-Branch: FG DTM-L1+L2 | 83.8 (4.57) | 72.3 (6.78) | 28.5 (14.1) | 7.47 (3.40) |
| Rec-Branch: SDF-L1 | 82.5 (9.05) | 73.6 (10.9) | 12.0 (4.61) | 2.73 (1.38) |
| Rec-Branch: SDF-L2 | **86.9 (4.43)** | **77.1 (7.92)** | **10.2 (6.03)** | **2.71 (1.68)** |
| Rec-Branch: SDF-L1+L2 | 85.1 (67.5) | 74.6 (9.24) | 16.7 (13.1) | 4.00 (3.19) |

### 4.1. Dataset 1: Left atrial MRI

Table 1 presents the quantitative results for left atrial MRI segmentation. Compared with the naive V-Net baseline, the two types of methods (New loss functions and adding auxiliary tasks) can obtain performance gains. Specifically, Hausdorff distance loss, multi-heads CNN and Rec-Branch CNN improved the baseline by 1.1%, 2.6%, and 2.5% in terms of Dice, respectively. SDF loss improved 95HD by 6.6, Multi-heads CNN and Rec-Branch CNN also improved 95HD by 8.3 and 9.9, respectively. Multi-heads CNN achieved the best Dice, Jaccard and ASD, and Rec-Branch CNN achieved the best 95HD with approximate 50% reduction. Paired T-test shows that the improvements are statistically significant at $p < 0.01$. We also found the regression branch and loss functions have significant impact on the performance. In particular, adding SDF regression task can provide better performance compared with adding the foreground DTM regression. It can be found that adding foreground distance map regression as an auxiliary task even degrades the performance

compared with baseline in both multi-heads and Rec-Branch CNNs. Moreover, using $L2$ loss is better than using $L1$ loss or their sum.

### 4.2. Dataset 2: Liver tumor CT

Table 2 shows the quantitative results on liver tumor CT dataset of the "winner" methods[6] in left atrial segmentation. Boundary loss and Hausdorff distance loss achieved minor improvements that are statistically significant at $p < 0.05$. It can be found that SDF loss, multi-heads and Rec-Branch CNNs didn't improve network performance. The potential reason may be that liver tumor segmentation is much more challenging than left atrial segmentation. For example, tumor has various location, shape and size, while these characteristics are relatively fixed for left atrial segmentation. It is non-trivial to regress the DTM or SDF of liver tumor.

Table 2: Quantitative results with average (standard deviation) on liver tumor CT dataset. The arrows indicate which direction is better.

| Methods | Dice ↑ | Jaccard ↑ | 95HD ↓ | ASD ↓ |
|---|---|---|---|---|
| V-Net baseline | 51.0 (28.8) | 39.8 (21.6) | 43.6 (45.2) | 14.9 (22.3) |
| Boundary loss | **52.5 (24.1)** | **41.0 (21.1)** | **26.3 (33.7)** | 7.70 (21.9) |
| Hausdorff distance loss | 52.0 (25.4) | 40.9 (22.6) | 28.8 (34.3) | **7.56 (19.4)** |
| Signed distance function loss | 47.6 (29.8) | 37.5 (26.9) | 31.1 (48.7) | 11.2 (23.8) |
| Multi-heads: SDF-L1 | 48.1 (27.6) | 38.2 (24.4) | 31.5 (40.6) | 8.11 (15.4) |
| Multi-heads: SDF-L2 | 47.1 (28.0) | 37.0 (25.3) | 25.5 (34.1) | 8.82 (22.3) |
| Rec-Branch: SDF-L1 | 48.4 (27.7) | 37.9 (25.3) | 32.2 (48.6) | 11.8 (31.1) |
| Rec-Branch: SDF-L2 | 48.6 (27.3) | 38.5 (25.0) | 31.0 (48.0) | 7.52 (21.8) |

## 5. Conclusion

For the question *"how can distance transform maps boost segmentation CNNs"*, our answer is that all the benchmark methods have the potential to improve the performance of baseline CNNs based on the experimental results. However, the performance gains are not consistent in different datasets. In particular, implementation details have remarkable effects on the final performance, for example learning rates, regression tasks, loss functions and so on. In practice, we would recommend multi-heads and Rec-Branch CNNs for the first try in organ segmentation tasks. On the other hand, boundary loss and Hausdorff distance loss would be suggested for the first try in tumor segmentation tasks. Importantly, how should we use the distance transform maps to boost existing CNNs and obtain **robust** performance gains is still an open question.

We can not claim we have completely reproduced the five benchmark methods, because most of them are not open-source except boundary loss[7]. However, we tried our best

---

6. "Winner" methods: the methods that achieve performance improvements.
7. https://github.com/LIVIAETS/surface-loss

to tune each method to achieve the best performance. For example, we tried different learning rates for each experiments. We also tried different $\alpha$ decay rates for boundary loss and Hausdorff distance loss. More than 70 experiments were run to ensure a fair comparison of these methods as shown in Appendix. Another limitation is that we used V-Net as backbone without justification. In future work, we will evaluate these methods with recent network architectures on more segmentation datasets, for example the Medical Segmentation Decathlon (Simpson et al., 2019). Furthermore, exploring the combination of the two different kinds of methods is also a promising extension. Our codes and trained models are publicly available at https://github.com/JunMa11/SegWithDistMap, which we hope would be useful for the community.

## Acknowledgments

This project is supported by the National Natural Science Foundation of China (No. 91630311, No. 11971229). The authors would also like to thank the organization team of MICCAI 2017 liver tumor segmentation challenge MICCAI 2018 and left atrial segmentation challenge for the publicly available dataset. We also thank the reviewers for their valuable comments and suggestions. Last but not least, we thank Lequan Yu for his great PyTorch implementation of V-Net (Yu et al., 2019) and Fabian Isensee for his great PyTorch implementation of U-Net (Isensee et al., 2020).

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

## Appendix A. Hyper-parameters trials for boundary loss

Table 3 and Table 4 present the hyper-parameters experiments of boundary loss.

Table 3: Boundary loss (BD) with different learning rates (LR), $\alpha$ decay rates and signed distance functions (non-normalized or normalized to $[-1, 1]$) on left atrial MRI dataset. Failed means the training does not converge.

| Methods | LR | $\alpha$ decay | Dice | Jaccard | ASD | 95HD |
|---------|-----|---------|------|---------|-----|------|
| BD | 0.001 | 0.01 | 0.766 | 0.643 | 15.758 | 3.884 |
| BD | 0.001 | 0.001 | 0.801 | 0.677 | 30.219 | 8.980 |
| BD | 0.0001 | 0.01 | 0.246 | 0.166 | 35.736 | 7.086 |
| BD | 0.0001 | 0.001 | 0.625 | 0.509 | 25.370 | 5.026 |
| BD Norm. SDF | 0.001 | 0.01 | 0.659 | 0.515 | 30.551 | 9.804 |
| BD Norm. SDF | 0.001 | 0.001 | 0.777 | 0.647 | 28.666 | 8.756 |
| BD Norm. SDF | 0.0001 | 0.01 | | | Failed | |
| BD Norm. SDF | 0.0001 | 0.001 | 0.460 | 0.338 | 31.945 | 9.350 |
| BD Norm. SDF | 0.01 | 0.001 | **0.850** | **0.742** | **20.823** | **5.435** |

Table 4: Boundary loss (BD) with different learning rates (LR), $\alpha$ decay rates and signed distance functions (non-normalized or normalized to $[-1, 1]$) on liver tumor CT dataset. Failed means the training does not converge.

| Methods | LR | $\alpha$ decay | Dice | Jaccard | ASD | 95HD |
|---------|-----|---------|------|---------|-----|------|
| BD | 0.001 | 0.01 | 36.470 | 27.130 | 30.770 | 3.860 |
| BD | 0.001 | 0.001 | 0.511 | 0.398 | 29.351 | 8.316 |
| BD | 0.0001 | 0.01 | | | Failed | |
| BD | 0.0001 | 0.001 | | | | |
| BD Norm. SDF | 0.001 | 0.01 | 0.4877 | 0.3784 | 38.460 | 16.036 |
| BD Norm. SDF | 0.001 | 0.001 | **0.525** | **0.410** | **26.317** | **7.698** |
| BD Norm. SDF | 0.0001 | 0.01 | | | Failed | |
| BD Norm. SDF | 0.0001 | 0.001 | | | | |

## Appendix B. Hyper-parameters trials for Hausdorff distance loss

Table 5 and Table 6 present the hyper-parameters experiments of Hausdorff distance loss.

## Appendix C. Hyper-parameters trials for signed distance function loss

Table 7 presents the hyper-parameters experimental results of signed distance function loss.

Table 5: Hausdorff distance loss (HD) with different learning rates (LR), $\alpha$ decay rates and distance transform map (non-normalized DTM or normalized DTM to $[0, 1]$) on left atrial MRI dataset.

| Methods | LR | $\alpha$ decay | Dice | Jaccard | ASD | 95HD |
|---|---|---|---|---|---|---|
| HD | 0.001 | 0.01 | 0.656 | 0.503 | 40.770 | 14.810 |
| HD | 0.001 | 0.001 | 0.757 | 0.623 | 27.640 | 7.625 |
| HD | 0.0001 | 0.01 | 0.723 | 0.578 | 37.630 | 12.600 |
| HD | 0.0001 | 0.001 | 0.640 | 0.485 | 40.050 | 14.360 |
| HD Norm. DTM | 0.001 | 0.01 | 0.474 | 0.335 | 40.940 | 14.080 |
| HD Norm. DTM | 0.001 | 0.001 | 0.773 | 0.641 | 31.020 | 9.765 |
| HD Norm. DTM | 0.0001 | 0.01 | 0.252 | 0.157 | 47.670 | 19.940 |
| HD Norm. DTM | 0.0001 | 0.001 | 0.400 | 0.276 | 38.860 | 14.140 |
| HD Norm. DTM | 0.01 | 0.001 | **0.855** | **0.750** | **15.921** | **4.461** |

Table 6: Hausdorff distance loss (HD) with different learning rates (LR), $\alpha$ decay rates and distance transform map (non-normalized DTM or normalized DTM to $[0, 1]$) on liver tumor CT dataset. Failed means the training does not converge.

| Methods | LR | $\alpha$ decay | Dice | Jaccard | ASD | 95HD |
|---|---|---|---|---|---|---|
| HD | 0.001 | 0.01 | 0.292 | 0.196 | 76.793 | 39.510 |
| HD | 0.001 | 0.001 | 0.519 | 0.405 | 34.884 | 11.152 |
| HD | 0.0001 | 0.01 | | | Failed | |
| HD | 0.0001 | 0.001 | 0.294 | 0.211 | 53.763 | 25.509 |
| HD Norm. DTM | 0.001 | 0.01 | 0.478 | 0.370 | 43.546 | 19.233 |
| HD Norm. DTM | 0.001 | 0.001 | 0.520 | 0.409 | 28.820 | 7.562 |
| HD Norm. DTM | 0.0001 | 0.01 | | | Failed | |
| HD Norm. DTM | 0.0001 | 0.001 | | | | |

Table 7: Signed distance function (SDF) loss ablation study results with different learning rates (LR) on left atrial dataset.

| Methods | LR | Dice | Jaccard | ASD | 95HD |
|---|---|---|---|---|---|
| Dice loss+L1 | 0.01 | 0.847 | 0.739 | 23.260 | 6.572 |
| Dice loss+L1 | 0.001 | 0.771 | 0.658 | 19.750 | 5.490 |
| Dice loss+L1+SDF loss | 0.01 | 0.813 | 0.704 | 16.090 | 4.044 |
| Dice loss+L1+SDF loss | **0.001** | **0.842** | **0.735** | **13.540** | **3.243** |

## Appendix D. Hyper-parameters trials for multi-heads V-Net

Table 8 and 9 present the hyper-parameters experimental results of multi-heads V-Net.

Table 8: Multi-heads V-Net with different regression tasks, loss functions and learning rates on left atrial dataset.

| Multi-heads | LR | Dice | Jaccard | ASD | 95HD |
|---|---|---|---|---|---|
| FG DTM regression-L1 | 0.01 | 0.837 | 0.725 | 24.676 | 6.622 |
| FG DTM regression-L1 | 0.001 | 0.837 | 0.725 | 23.712 | 6.226 |
| FG DTM regression-L2 | 0.01 | 0.798 | 0.671 | 14.504 | 3.076 |
| FG DTM regression-L2 | 0.001 | 0.826 | 0.709 | 15.564 | 4.101 |
| FG DTM regression-L1+L2 | 0.01 | 0.814 | 0.695 | 19.087 | 4.919 |
| FG DTM regression-L1+L2 | 0.001 | 0.833 | 0.726 | 17.452 | 4.867 |
| SDF regression-L1 | 0.01 | 0.855 | 0.753 | **11.823** | **2.646** |
| SDF regression-L1 | 0.001 | 0.817 | 0.703 | 17.632 | 4.044 |
| SDF regression-L2 | 0.01 | **0.870** | **0.772** | 16.119 | 3.970 |
| SDF regression-L2 | 0.001 | 0.772 | 0.657 | 28.987 | 6.609 |
| SDF regression-L1+L2 | 0.01 | 0.845 | 0.734 | 24.713 | 6.093 |
| SDF regression-L1+L2 | 0.001 | 0.796 | 0.691 | 17.217 | 4.315 |

Table 9: Multi-heads V-Net (signed distance function regression) with different loss functions and learning rates (LR) on liver tumor CT dataset.

| Multi-heads | LR | Dice | Jaccard | ASD | 95HD |
|---|---|---|---|---|---|
| SDF regression-L1 | 0.01 | 0.4841 | 0.3819 | 31.5352 | 8.1127 |
| SDF regression-L1 | 0.001 | 0.4705 | 0.3718 | 31.6681 | 8.4459 |
| SDF regression-L2 | 0.01 | 0.4672 | 0.3649 | 30.7485 | 9.8766 |
| SDF regression-L2 | 0.001 | 0.471 | 0.3704 | 25.4891 | 8.8161 |

## Appendix E. Hyper-parameters trials for reconstruction-branch V-Net

Table 10 and 11 present the hyper-parameters experimental results of reconstruction-branch V-Net.

Table 10: Reconstruction-branch V-Net with different regression tasks, loss functions and learning rates on left atrial dataset.

| Rec-Branch | LR | Dice | Jaccard | ASD | 95HD |
|---|---|---|---|---|---|
| FG DTM regression-L1 | 0.01 | 0.835 | 0.722 | 23.552 | 5.450 |
| FG DTM regression-L1 | 0.001 | 0.830 | 0.715 | 26.234 | 6.997 |
| FG DTM regression-L2 | 0.01 | 0.798 | 0.672 | 24.431 | 6.932 |
| FG DTM regression-L2 | 0.001 | 0.815 | 0.695 | 19.484 | 4.488 |
| FG DTM regression-L1 + L2 | 0.01 | 0.774 | 0.638 | 23.541 | 6.531 |
| FG DTM regression-L1 + L2 | 0.001 | 0.838 | 0.723 | 28.466 | 7.466 |
| SDF regression-L1 | 0.01 | 0.843 | 0.737 | 12.007 | 2.734 |
| SDF regression-L1 | 0.001 | 0.811 | 0.694 | 18.274 | 4.508 |
| SDF regression-L2 | 0.01 | **0.869** | **0.771** | **10.234** | **2.714** |
| SDF regression-L2 | 0.001 | 0.800 | 0.686 | 19.129 | 4.830 |
| SDF regression-L1 + L2 | 0.01 | 0.851 | 0.746 | 16.672 | 4.003 |
| SDF regression-L1 + L2 | 0.001 | 0.820 | 0.704 | 15.254 | 3.284 |

Table 11: Multi-Head V-Net (signed distance function regression) with different loss functions and learning rates (LR) on liver tumor CT dataset.

| Rec-Branch | LR | Dice | Jaccard | ASD | 95HD |
|---|---|---|---|---|---|
| SDF regression-L1 | 0.01 | 0.484 | 0.379 | 32.249 | 11.786 |
| SDF regression-L1 | 0.001 | 0.467 | 0.366 | 32.844 | 6.687 |
| SDF regression-L2 | 0.01 | 0.447 | 0.343 | 42.535 | 15.428 |
| SDF regression-L2 | 0.001 | 0.486 | 0.385 | 30.996 | 7.550 |
| SDF regression-L1+L2 | 0.01 | 0.429 | 0.333 | 42.095 | 14.791 |
| SDF regression-L1+L2 | 0.001 | 0.456 | 0.353 | 34.837 | 8.522 |

