# OpenReview forum: "How Distance Transform Maps Boost Segmentation CNNs: An Empirical Study"
_MIDL.io/2020/Conference — MIDL 2020_

### Official Review · AnonReviewer3 · 2020-03-09
**A good review and analysis of distance maps in image segmentation**

**Rating:** 4
**Confidence:** 5
**Recommendation:** Poster

**Summary:**

This paper proposes to analyze various ways of embedding distance maps (DM) to improve deep-learning based image segmentation. The authors compare several approaches of the literature using two segmentation datasets and confirm that it is a useful information to add during training. Results seem to suggest that it is preferable to learn to generate the DM rather than to use DM in an additional cost function. Moreover the authors show that regressing the signed distance function is preferable over generating a distance transform map.

**Strengths:**

 The paper is well written and easy to follow. In my opinion it is a good match as it is an important message to convey to the community that, in general, adding distance-based information improves segmentation results.

**Weaknesses:**

The authors have performed experiments on only two datasets. Using more open segmentation datasets such as for example those from the medical image decathlon would have strengthened the significance of the conclusions drawn.


**Detailed Comments:**

The authors argue that it is the first paper proposing to regress SDM rather than DTM, but the distinction between the two is unclear to me, is it only a matter of sign ? If so, how could that possibly explain the performance gains ? Please provide a few additional words to discuss this.

**Justification Of Rating:**

This is a simple and clear study on the benefits of distance maps in deep-learning based segmentation. Because the conclusions drawn are very general, this is I believe a good match for a conference such as MIDL. I understand it is not a major methodological contribution, but it is a simple and clear message to convey to the community.

**Paper Type:**

both

**Special Issue:**

no

---

> ### Author Response · Authors · 2020-03-27
> **Formal definition of distance transform map and signed distance function.**
>
> Thank you for your time and valuable comments. Your question is important to help us clarify the basic terms in this paper.
>
> > Using more open segmentation datasets such as for example those from the medical image decathlon would have strengthened the significance of the conclusions drawn.
>
> - We agree that more experiments are necessary to enhance the conclusion in this work. We would like to address this limitation more extensively in future work.
>
> > The authors argue that it is the first paper proposing to regress SDM rather than DTM, but the distinction between the two is unclear to me, is it only a matter of sign ? If so, how could that possibly explain the performance gains ? Please provide a few additional words to discuss this.
>
> - we added a new subsection (2.1) to formally define the distance transform map (DTM) and the signed distance function (SDF) of the ground truth $G$ by
> \begin{equation}\label{Eq:DTM}
>   G_{DTM} =
> \begin{cases}
>     \inf\limits_{y\in \partial G}||x-y||_2, & x \in G_{in}  \\
>     0, & others
> \end{cases}
> \end{equation}
> where $||x-y||_2$ is the Euclidian distance between voxels $x$ and $y$, and $G_{in}$ denotes the inside of the object.
> The signed distance function (SDF) of ground truth $G$ is defined by
> \begin{equation}\label{Eq:LSF}
>   G_{SDF} =
> \begin{cases}
>     -\inf\limits_{y\in \partial G}||x-y||_2, & x \in G_{in} \\
>     0, & x \in \partial G \\
>     \inf\limits_{y\in \partial G}||x-y||_2, & x \in G_{out}
> \end{cases}
> \end{equation}
> where $G_{out}$ and $\partial G$ denote the outside and boundary of the object, respectively.
>
> The main difference between them is that the SDF considers the distance transformation information of both foreground and background, while DTM only computes the distance transformation of the foreground.

---

### Official Review · AnonReviewer2 · 2020-03-13
**Comparative evaluations of DTM-based techniques for segmentation with two public datasets**

**Rating:** 4
**Confidence:** 4
**Recommendation:** Oral

**Summary:**

This work summary the latest developments of incorporating of distance transform maps (DTM) of ground truth into segmentation CNNs, and evaluation results of five benchmark methods on two typical public datasets. As a summary, the authors divided the five methods into two categories: new loss functions with distance transform maps and additional auxiliary tasks with distance transform maps. They then evaluated these five methods with left atrial MRI and liver tumor CT datasets. The conclusion is that incorporating DTMto segmentation can improve performance, but not always in some methods.

**Strengths:**

Comparative evaluations of parts of the-state-of-the-art methods on well-known public dataset are presented. These results help to develop new methodologies and application systems. Especially, all most of the selected five benchmark methods are not open source. Furthermore, authors presented results of the best hyperparameter search as appendix.

**Weaknesses:**

No theoretical insights to the results of comparisons. What lead the different results in two datasets is still unclear for readers, even though I know this requirement is difficult for deep-learning-based approaches.

The selection criteria of methods and survey style are both unclear.

**Detailed Comments:**




**Justification Of Rating:**

Reporting of comparative evaluations of the-state-of-the-methods of incorporating distance transform maps to segmentation CNN in different two public datasets is worthy for presentation. Especially, it includes non-open-source new methods.

**Paper Type:**

validation/application paper

**Questions To Address In The Rebuttal:**

I found works, which is not mentioned in this manuscript, but are relevant works. For an example, I pick up the following MICCAI paper.
https://link.springer.com/chapter/10.1007%2F978-3-030-32226-7_39

How did you check the relevant methods and select them for comparison? I am not sure this submission fully covered the state-of-the-art methods.


**Special Issue:**

no

---

> ### Author Response · Authors · 2020-03-27
> **Selection criteria of methods**
>
> Thank you for your time and effort in reviewing our paper. Your question is valuable to help us further clarify our motivation.
>
> > How did you check the relevant methods and select them for comparison? I am not sure this submission fully covered the state-of-the-art methods.
>
> We selected these methods based on the following criteria:
> - The method is general and can be applied to many 3D segmentation tasks.
> - The method is published in 2019-2020.
> We also clarified the criteria in the revised manuscript.
>
> The MICCAI you mentioned is tailored for vessel segmentation as the author said in the abstract "we present a novel loss function, coined radial distance loss, specifically designed for tubular structures. "
>
> To ensure our work covers the SOTA methods, we searched several well-known medical image analysis publications (e.g., MedIA, TMI, MICCAI) by the keywords segmentation AND distance

---

### Official Review · AnonReviewer1 · 2020-03-13
**Review of How Distance Transform Maps Boost Segmentation CNNs: An Empirical Study**

**Rating:** 2
**Confidence:** 5

**Summary:**

The authors summarize current findings about the incorporation of distance maps to the training of segmentation network. They also benchmark five methods in two datasets.
The analysis of results and harmonizing of methods is imperfect and, as this work is a benchmark study, there is no technical novelty. I would have expected a finer analysis of the results for such a contribution.


**Strengths:**

-Very relevant research topic. Try to give insights on general improvement of segmentation with CNNs.
-willingness to make code open-source


**Weaknesses:**

-they methods could have been presented in a more harmonized fashion
-the quantitative analysis lacks precision (see detailed comments)
-no qualitative analysis
-the results are somehow inconclusive


**Detailed Comments:**

-In equation (2), the symbol for voxel-wise multiplication is not defined. And other symbols for the same operation are used elsewhere
-Section 2.2 is wordy and redundant. The same information could be given using significantly less place.
-No standard deviation, confidence interval or statistical test reported in the results. So in most cases it is difficult to conclude.

Minor comments:
-”standard variance.”? Variance or Standard deviation?



**Justification Of Rating:**

The improvement of the methods is too small to have a practical impact. The paper does not give new methodological insights.  The comparison between methods could be harmonized better.

**Paper Type:**

methodological development

**Questions To Address In The Rebuttal:**

-The authors should harmonize the notations and some of the notations do not make sense. For example, equation (1) and (2). In (1), an integral was used to span omega, and in (2), a sum was used. Chose one. According to the definition of omega in the manuscript, it should be integral. But because you work with numerical images, it does not make sense to define a continuous space. So I would adapt to definition of omega and use sums everywhere.
-I do not think the cleavage of methods between new loss functions and auxiliary tasks is very relevant. Especially considering that each of these now loss functions need to be added the standard dice loss during training. In all studied methods, an auxiliary task based on the distance map is added. The auxiliary task is incorporated to the optimization by changing the definition of loss function. So for all methods, both a new loss function and an auxiliary task are introduced.

**Special Issue:**

no

---

> ### Author Response · Authors · 2020-03-27
> **Harmonize the notations and clarify the classification criteria**
>
> We thank the reviewer for the detailed comments and useful suggestions provided to improve the manuscript.
>
> > The authors should harmonize the notations and some of the notations do not make sense. In equation (2), the symbol for voxel-wise multiplication is not defined. And other symbols for the same operation are used elsewhere
>
> - We added a subsection to formally define the distance transform map and signed distance function at the beginning of Section 2 (refer to the reply to the 4th reviewer).
> We also unified the notations by defining \Omega as the image grid, and  $\circ$ is the Hadamard (i.e. voxel-wise) product.
>
>
> > I do not think the cleavage of methods between new loss functions and auxiliary tasks is very relevant
>
> We appreciate that the reviewer made a very important observation.
> We agree that these two categories (new loss functions and adding auxiliary tasks) have similarities since both of them involve changes to loss functions.
> However, the second type of methods focused on regression as an auxiliary task and added naive losses such as L1, L2, while the first type of method aimed to design novel loss functions.
>
> In addition, your question inspires us to explore the combination of the two different methods that we would like to address in future work.
>
> > Section 2.2 is wordy and redundant. The same information could be given using significantly less place.
> Thanks for the great suggestion. To make this section more concise, we rewrote it, and the paragraph length was halved.
>
> > No standard deviation, confidence interval or statistical test reported in the results. So in most cases it is difficult to conclude.
>
> We added standard deviation and paired T-test results to the quantitative results (p<0.05), and our primary conclusions can still hold.
>
> Typos are fixed.

---

### Official Review · AnonReviewer4 · 2020-03-14
**An experimental comparison study of five methods for incorporating distance transform maps of ground truth into segmentation CNNs for medical image analysis**

**Rating:** 3
**Confidence:** 4
**Recommendation:** Poster

**Summary:**

This paper presents an experimental comparison study of five methods for incorporating distance transform maps of ground truth into segmentation CNNs for medical image analysis. The V-Net is used as baseline. The test is done on two datatsets. The overall picture turns out to be mixed: There is no consistent performance enhancement when using the distance transform maps. In addition, the implementation details have remarkable effects on the final performance.

**Strengths:**

The experimental work here is overall well done (although with some limitation, see below).

The paper is well clearly structured and easy to read. Such details like the arrows in the caption of Tables indicating which direction is better is a real help.

The authors give a realistic picture of the gains and limitations of the studied methods.

The code, trained models and training logs will be publicly available.


**Weaknesses:**

Such experimental studies are typically somewhat limited. This is also the case here. Only V-net is used as baseline, which is also not justified at all. The question arises to which extent the findings here generalize to other nets. Certainly, the same applies to the tasks with the related datatsets.

**Detailed Comments:**

“In practice, it should be noted that the three loss functions should be coupled with Dice loss so as to stabilize training process, especially at the beginning of the training. Otherwise the training may not converge”. More details will be helpful to have a better understanding and support for potential users.

Some minors: “SDF loss do not improve”, “The other is the the liver tumor” (this may not be complete, please check).



**Justification Of Rating:**

This experimental study, although with limitations, does have some value to the research community. It will further increase the interest in investigating incorporating distance transform maps of ground truth into segmentation CNNs, in particular from a methodological perspective. In addition, the authors will make the code, trained models and training logs available, which will further support the use of such methods by other researchers.

**Paper Type:**

validation/application paper

**Special Issue:**

no

---

> ### Author Response · Authors · 2020-03-27
> **Details about the usage of three loss functions**
>
> We thank the reviewer for the care with which the baseline and the usage details of the loss functions.
>
> > Only V-net is used as baseline, which is also not justified at all. The question arises to which extent the findings here generalize to other nets. Certainly, the same applies to the tasks with the related datasets.
>
> - We agree that evaluating the five methods with more network architectures is important to enhance the conclusion in this work. We will address this limitation in the extension of this work.
>
>
> > More details will be helpful to have a better understanding and support for potential users.
>
> - We introduced the details in Section 3.2, these loss functions should be used by
> \begin{equation*}
>   L = \alpha L_{Dice} + (1-\alpha)L_{(\cdot)},
> \end{equation*}
> where $(\cdot)$ denotes the new losses, and $\alpha>0$ is the weight hyper-parameter. For example, the weight $\alpha$ can be initially set to 1, and decrease by 0.01 after each epoch until it reaches the value of 0.01.
>
>
> > Some minors: “SDF loss do not improve”, “The other is the the liver tumor” (this may not be complete, please check).
>
> - Typos are fixed.

---

### Meta-Review · Area_Chair1 · 2020-04-06
**MetaReview of Paper2 by AreaChair1**

**Rating:** 4
**Recommendation For Accepted Papers:** Poster

**Metareview:**

The paper’s strengths have been praised unanimously by the reviewers:

- Authors address a hot topic on how to improve medical image segmentation with distance transform maps.
- The paper is well written and easy to follow with a clear take-home message  (and limitations are acknowledged)
- Results have been assessed on well-known public dataset are presented.
- Code and models will be publicly released

While reviewers had somehow identified the same strengths, the weaknesses are different depending to the reviewers, and not critical: limited number of datasets or models (only V-net), the selection criteria of methods for the survey, no theoretical insights to the results of comparisons - although this last is acknowledged as difficult to establish.

The reviewer who was in favor of a weak reject noted that the results were somehow inconclusive. The authors replied that statistical tests had now been performed to support the results.

Now that this issue has been addressed, I believe this paper can accepted.

**Paper Type:**

both

**Special Issue:**

no

---

### Decision · Program_Chairs · 2020-04-11

Accept